# Moving in the Dark—Evidence for an Influence of Artificial Light at Night on the Movement Behaviour of European Hedgehogs (*Erinaceus europaeus*)

**DOI:** 10.3390/ani10081306

**Published:** 2020-07-30

**Authors:** Anne Berger, Briseida Lozano, Leon M. F. Barthel, Nadine Schubert

**Affiliations:** 1Department of Evolutionary Ecology, Leibniz-Institute for Zoo and Wildlife Research, Alfred-Kowalke-Straße 17, 10315 Berlin, Germany; berger@izw-berlin.de (A.B.); briseida.lozano@gmx.de (B.L.); Barthel.leon@gmx.de (L.M.F.B.); 2Institut für Ökologie, Technische Universität Berlin, Rothenburgstraße 12, 12165 Berlin, Germany; 3Department of Animal Behaviour, Bielefeld University, Konsequenz 45, 33615 Bielefeld, Germany

**Keywords:** hedgehogs, *Erinaceus europaeus*, light pollution, ALAN, GPS, acceleration, activity, movement behaviour, urbanisation, conservation

## Abstract

**Simple Summary:**

The European hedgehog is one of the most popular and well-known wild animals, but its numbers are declining throughout Europe, especially in rural areas. Effective hedgehog conservation requires an understanding of the hedgehog’s ability to adapt to a changing environment. Due to globally increasing urbanisation, the use of artificial light sources to illuminate the night, called light pollution, has spread dramatically. Light pollution significantly affects the behaviour and ecology of wildlife, but the hedgehog’s behaviour towards light pollution remains unknown. We therefore investigated the effects of light pollution on the natural movement behaviour of hedgehogs living in an urban environment. Although hedgehogs can react very variably to environmental influences, the majority of hedgehogs studied here preferred to move in less illuminated rather than in strongly illuminated areas. This apparently rigid behaviour could be used in applied hedgehog conservation to connect isolated hedgehog populations or to safely guide the animals around places dangerous for them via dark corridors that are attractive for hedgehogs.

**Abstract:**

With urban areas growing worldwide comes an increase in artificial light at night (ALAN), causing a significant impact on wildlife behaviour and its ecological relationships. The effects of ALAN on nocturnal and protected European hedgehogs (*Erinaceus europaeus*) are unknown but their identification is important for sustainable species conservation and management. In a pilot study, we investigated the influence of ALAN on the natural movement behaviour of 22 hedgehogs (nine females, 13 males) in urban environments. Over the course of four years, we equipped hedgehogs at three different study locations in Berlin with biologgers to record their behaviour for several weeks. We used Global Positioning System (GPS) tags to monitor their spatial behaviour, very high-frequency (VHF) loggers to locate their nests during daytime, and accelerometers to distinguish between active and passive behaviours. We compared the mean light intensity of the locations recorded when the hedgehogs were active with the mean light intensity of simulated locations randomly distributed in the individual’s home range. We were able to show that the ALAN intensity of the hedgehogs’ habitations was significantly lower compared to the simulated values, regardless of the animal’s sex. This ALAN-related avoidance in the movement behaviour can be used for applied hedgehog conservation.

## 1. Introduction

The European hedgehog (*Erinaceus europaeus*) is a solitary, hibernating, nocturnal insectivore that is one of the most popular and well-known wild species. Recent studies show that population densities of hedgehogs in cities and suburbs are higher than those in the countryside [1,2,3]. Moreover, long-term monitoring studies found that the overall hedgehog population in various countries is declining, in some places dramatically [4]. Consequently, in some cases the hedgehog’s protection status has been upgraded. In Great Britain, for example, the population has declined from about 1.5 million hedgehogs in 1995 to 522,000 in 2016 [5]. In Germany, only two long-term monitoring studies on a very local scale exist, showing similar decreases to those in the UK [6,7], and concerns about the decline in hedgehogs have also been expressed in other European countries [1,8,9]. The underlying mechanisms causing these declines are certainly complex and multifactorial, including habitat loss, interspecific competition, road collisions, and the intensification of agriculture [4,10,11,12,13,14]. Due to increasing fragmentation and decreasing density of hedgehog populations, the danger of the formation of island populations and inbreeding is already being discussed [15,16,17,18]. 

In order to develop effective protection concepts, threats to hedgehogs as well as their limits of adaptation must be determined precisely. Although the hedgehog’s biology and care husbandry are well known, there is—due to its cryptic lifestyle—hardly any knowledge about its behaviour in the wild, the influence of environmental factors on its behaviour, and its adaptation limits. 

With the steadily growing human population and the increasing urbanisation worldwide, the amount and intensity of artificial light at night (ALAN) is increasing as well [19]. ALAN has dramatic ecological effects and causes considerable impacts on the migratory behaviour [20], reproduction [21,22], fitness [23], predator-prey interaction [24], activity [25] or phenology [26] of certain species. However, the effects of ALAN on mammalian species other than bats have been studied very little [27,28]. Thus far, one study on the effect of ALAN on hedgehogs’ activity at supplementary feeding stations has been carried out [29] and did not find a significant effect. As mentioned by the authors, the drive to obtain high-energy resources at the illuminated feeding stations may outweigh the hedgehogs’ light-avoidance behaviour. Thus, this study could not clarify the natural reaction of hedgehogs to ALAN [29].

In our study we investigate to what extent ALAN influences the movement behaviour of wild hedgehogs inhabiting urban spaces. With the rising numbers of hedgehogs living in urban environments, as well as increasing light pollution, it is important to determine the influence of ALAN on this protected species. If hedgehogs show any preference or avoidance, guiding systems for hedgehogs could be developed to safely lead them around risky places (e.g., busy roads) or to link fragmented populations. We have concentrated on the movement behaviour of hedgehogs, as we suspect the greatest opportunity for behavioural adaptability here: hedgehogs that are not moving are either in their nests, which are usually situated in (light-)protected vegetation or constructs, or they are hunting natural food in places where it is available [30,31]. However, the most common natural prey items of hedgehogs [32,33,34] differ in their behaviour towards light: whether earthworms (Lumbricidae) can perceive light is unknown, but ground beetles (Carabidae) are more abundant in artificially lit compared to dark areas [35], and woodlice (Oniscidea) are light averse. This availability and distribution of prey will in turn affect the movement behaviour of hedgehogs to a certain extent.

In this study we aimed to test the effect of ALAN on the natural movement behaviour of 22 hedgehogs (nine females and 13 males) living in urban environments by tagging their movement using Global Positioning System (GPS) loggers and acceleration data.

## 2. Materials and Methods

### 2.1. Study Area

The study was conducted in three different areas in Berlin, the capital of Germany, which has about 4 million inhabitants: in Tierpark Berlin–Friedrichsfelde in the east of Berlin (52°30′11.7′′ N, 13°31′47.1′′ E), in Treptower Park in the southeast of the city (52°29′18.5′′ N, 13°28′11.1′′ E) and around the S-Bahn station Tegel in the North (52°35′18.0′′ N, 13°17′23.0′′ E) (Figure 1). All areas contain green spaces (meadows, bushes, hedges, and large trees), playgrounds, larger sealed areas, and footpaths. 

The Tierpark (TP) is a zoological garden of about 160 ha in size. It is open to the public daily from 09:00 to 18:30 CET and dogs are allowed on a leash. Apart from the security service, the TP is free of people and traffic at night. The TP contains numerous animal enclosures, creating a mosaic-like fragmented habitat with many areas inaccessible to hedgehogs. 

The Treptower Park (TR) is a public city park of 88.2 ha in size. It is accessible from all sides, open 24 h a day, and is used a lot by citizens for recreation, especially for picnics, sports games or for walking dogs.

The area of the study location in Tegel (TE) is about 52 ha in size and consists mainly of a residential area with public roads and some industrial locations (such as railway facilities), containing multi-floor blocks of houses as well as detached houses surrounded by gardens. Between these houses there are streets and paths that are always open to the public. 

The TP and the TR do not contain residential buildings or lights as a source of ALAN. However, both parks are at some sides bordered by big streets that are strongly illuminated at night. In TE, all roads and railway facilities are continuously illuminated at night. 

### 2.2. Field Work

The fieldwork took place from 10th of August to 19th of September 2016 (TR), from 14th of August to 4th of September 2017 (TP), from 22nd of May to 30th of June 2018 (TP) and from 20th of July to 8th of October 2019 (TE).

At the beginning of each field season, several nightly searches (starting around one hour after sunset) were carried out in the corresponding study area to find active hedgehogs using flashlights (P14.2, LED Lenser, Solingen, Germany). Each hedgehog found was weighed, sexed and marked with five shrink tubes glued on the spines [36]. The markings were of different colour in each study year and were given an uprising number starting with 1 to clearly identify each hedgehog in case of subsequent recaptures [37]. We equipped healthy hedgehogs with a body mass of more than 600 g with GPS-acceleration (ACC) loggers (E-obs GmbH, Munich, Germany) and very high-frequency (VHF) transmitters (Dessau Telemetry, Dessau, Germany), which we mounted on a backplate system glued to the hedgehogs’ spines [38].

In order to disturb the hedgehogs as little as possible, the tagged individuals were located every day during their stay in the daytime nest by means of the mounted VHF transmitter and a receiver (TRX-1000S, Wildlife Materials Inc., Murphysboro, IL, USA, or Wide Range Receiver AR 8200, AOR Ltd., Tokyo, Japan), and the exact GPS position of the nest was recorded using a Garmin GPSmap 60CSx device (Garmin Deutschland GmbH, Garching, Germany). At night, we only checked, by VHF distance tracking, whether the tagged individuals were moving (away from their nests). Every 4–5 nights, the tagged hedgehogs were located, caught, weighed and checked for potential problems with their backplate or their general health; these opportunities were also used for the necessary recharging of the GPS/ACC data loggers and the GPS/ACC data download.

At the end of the studies, the animals were caught again, weighed, the loggers were removed, and the backplate system was cut off the spines.

All procedures performed involving handling of animals were in accordance with the ethical standards of the institution (IZW permit 2016-02-01) and German federal law (permission number Reg0115/15 and G0104/14).

### 2.3. Data Logger Setup 

In order to be able to find the tagged individuals at any time, the VHF transmitters were programmed to send signals continuously. In addition, GPS positions were recorded during the expected activity times of the nocturnal hedgehogs from 07 p.m. to 07 a.m. in 5-min intervals (TE_2019, TP_2018) or in 10-min intervals (TR_2016, TP_2017). This measurement was done in five-point shifts of one second difference. In addition to the GPS data, three-dimensional acceleration data were recorded. Here, a short burst of high-resolution data was recorded on all 3 axes at the same time. For the present study, the sampling frequency was 100 Hz per axis, a burst lasted 2.5 s and the bursts were recorded every minute of the day.

### 2.4. Data Analysis and Statistics

The data analysis and presentation were performed using R Studio version 3.5.1 software [39,40] and QGIS version 3.4.4-Madeira [41]. 

GPS locations that were recorded at times during which the data loggers were not attached to the animals, for example during transport of the loggers for attachment and detachment or during handling of the animals, as well as GPS locations identified as outliers (movement speed > 2 m/s were removed from the data set. Mean values of the GPS data collected within 10 s were calculated and then used for further analyses.

Individual GPS error (GPS_err) was determined by calculating the mean deviation of the GPS positions at times during which the hedgehog was immobile in its daily nest (07 a.m. to 07 p.m.) from the exact position of that nest, which we measured in the field.

Before calculating the home range (HR) with the cleaned GPS dataset (GPS_total), a bootstrap method and a site fidelity test were performed. The bootstrap method shows whether the number of locations is sufficient for calculating the home range, which is indicated by the curve reaching a plateau. HRs were calculated for those individuals that showed site fidelity by calculating the Minimum Convex Polygon (MCP100) for each hedgehog, using the standard bandwidth href for smoothing [42]. The principle of site fidelity means that only an animal whose occupied area is smaller than the estimated area occupied when the animal moves randomly shows site fidelity [43]. Spencer et al. [44] evaluated the site fidelity as a prerequisite for the presence of a home range in an animal. 

Afterwards, the GPS_total (measured between 07 p.m. and 07 a.m.) data set was divided into two groups: locations at times the hedgehog was considered to be moving (GPS_act) and at times the hedgehog was considered to be immobile (GPS_pas). As hedgehogs are strongly nocturnal, we expected them to have left the nest 1 h after sunset and to have returned to the nest 1 h before sunrise. Thus, we sorted all GPS locations recorded from 07 p.m. to 1 h after sunset and from 1 h before sunrise until 07 a.m. into GPS_pas as well to ensure that we only considered GPS locations for further analysis that had been recorded outside of the nest. For data collected between 1 h after sunset and 1 h before sunrise, we detected passive phases based on an acceleration data threshold (ACC_thres). We estimated this threshold individually for each hedgehog using the summed standard deviations (of the *x*, *y* and *z* axis of each burst, measured every minute) of the acceleration data. This acceleration threshold is expected to separate active and passive behaviours [45]. Locations recorded during this potential active phase of the hedgehogs (between 1 h after sunset and 1 h before sunrise) for which acceleration values dropped below ACC_thres were assigned to the GPS_pas dataset as well, as the hedgehog was considered immobile and we aimed to investigate movement behaviour. 

For the following calculations, we used the publicly available light intensity map of Berlin (resolution 1 sqm) [46] and the GPS locations of active (moving) hedgehogs (GPS_act). In the light map, light intensities measured via flyover are mapped using a relative scale from 0 (absolute darkness) to 48,429 (highest light intensity mapped). Thus, no measurement unit is given and increasing brightness of grey shades corresponds to increasing light intensities in the area (please see [46] for further information on the establishment and properties of the light map). A buffer (radius = mean GPS error) was placed around each of the GPS locations, light intensities were extracted and the mean light intensity of the area of these locations was calculated. Random GPS points with the same buffer size and the same number as the corresponding real measured locations were simulated within the individual’s MCP100 and light intensities were calculated in the same way as done for the actual GPS locations.

The mean light intensity of the real hedgehog’s locations and that of the random points were then used for comparison via the randomisation method. Utilising Equation (1) below, *p*-values for each hedgehog were established.
*p* = (sum(mean H0 ≤ meanobs) + 1) k + 1(1)
where mean H0: mean light intensity for simulated random points of individual study animal; meanobs: mean light intensity of individual study animal; k: number of repeats (=1000).

The obtained *p*-values were then analysed for all hedgehogs combined using Fisher’s method to investigate the behaviour towards ALAN for all hedgehogs together. The calculation of the combined *p*-values for all male hedgehogs was carried out in R Studio using the metap package [47].

## 3. Results

Data were collected successfully for 22 hedgehogs, including nine females and 13 males (Table 1). Over a period of 14–72 days, interspersed by regular breaks for the recharging of the loggers or due to technical issues with the backplate, we tracked the hedgehogs’ spatio-temporal behaviour using GPS tags and three-dimensional accelerometers and recorded between 487 and 2469 GPS locations per animal. Within this timespan, movements were tracked for most animals at least 21 days per individual, except for three animals (TR_08_2016, TR_09_2016 and TP_30_2018). For these hedgehogs, measurements were aborted ahead of schedule due to technical problems with the loggers.

The bootstrap method showed that we clearly exceeded the minimum number of GPS locations per animal necessary to calculate a reliable HR for all tagged hedgehogs (Appendix A). Site fidelity was detected for all 22 hedgehogs; however, eight animals, two females and six males (TR_09_2016, TP_13_2018, TP_21_2018, TP_26_2017, TE_02_2019, TE_03_2019, TE_05_2019, TE_07_2019), exhibited site fidelity not in both criteria, but only with regard to the mean square distance from the activity centre, the main criterion for site fidelity (Appendix A).

According to the individual activity thresholds (ACC_thres), the GPS locations were divided into locations recorded during inactive (GPS_pas) and active behaviour (GPS_act) (Table 1). With the exception of one female (TE_02_2019), who did not leave the nest for a few nights during the study period, all hedgehogs showed predominantly active behaviour during their nocturnal activity time from one hour after sunset until one hour before sunrise. Exemplarily, all buffered GPS locations (GPS_pas = yellow, GPS_act = red) are pictured on a map for one female and one male of each study area (Figure 2, Figure 3 and Figure 4). Blue spots within the individual HR represent the buffered random positions (its number corresponds to the number of the individual’s GPS_act locations). As an example, only a single simulation of the random points is indicated, but this simulation was repeated 1000 times for further analysis using the randomisation test. In the maps, the occurrence and intensity of ALAN is marked by grey shades, with increasing brightness corresponding to areas with higher ALAN intensity in the maps (Figure 2, Figure 3 and Figure 4).

Table 2 shows the light intensities of the buffered GSP locations measured for each individual (Animal_) and the corresponding value for the buffered random sites (Random_). The simulation of these random locations was repeated 1000 times per animal. The total of all random mean values of light intensity differed slightly between the study areas and ranges from 120.1 ± 77.9 (TP) over 122.2 ± 82.8 (TE) up to 124.8 ± 143.9 (TR) (Table 2). Five out of 22 hedgehogs did not show a mean light intensity at their locations that differed significantly from the randomly distributed locations in their MCP100. Out of these five individuals showing no statistically significant difference in the light intensity comparison, three animals (TE_03_2019, TE_05_2019 and TE_07_2019) exhibited mean light intensity values for their measured GPS locations (animal_mean) that were above the mean value determined for the randomly distributed locations (random_mean). These three individuals were also observed to have their nests or their movement paths situated along small bushes directly bordering the roadside or the track bed. The remaining 17 hedgehogs exhibited a significantly lower mean light intensity compared to that obtained via simulation of random locations (Table 2).

The difference between the random and the animals’ mean light intensities was compared between the sexes and study locations. The difference in the light intensities did not vary significantly between the sexes (Mann–Whitney U-Test, difference female = 15.2 ± 6.3, difference male = 5.9 ± 21.4, *p* = 0.5). Comparison of the differences between simulated and experienced light intensities of the hedgehogs in the different study locations (TR, TP and TE) yielded statistically significant differences (Kruskal Wallis rank sum test, *p* = 0.0005). Pairwise comparison revealed that the light intensity difference varied significantly between hedgehogs of TE and TR (Wilcoxon pairwise rank sum exact test, difference TE = −11.1 ± 24.1, difference TR = 20.2 ± 4.6, *p* = 0.007) and the hedgehogs of TE and TP (Wilcoxon pairwise rank sum exact test, difference TE = −11.1 ± 24.1, difference TP = 14.8 ± 4.5, *p* = 0.004).

## 4. Discussion

A comparison of the light intensity of each animal’s GPS locations extracted from a light map of Berlin with a corresponding number of simulated random locations within the animal’s HR indicates a preference for movement in locations with lower levels of ALAN compared to the simulated values. Our study is the first to observe such a preference in the movement behaviour of wild European hedgehogs in an unchanged, urban setting.

As ALAN is mostly emitted by streetlamps, it usually occurs locally (spotlike or linelike) with high intensity and especially in the vicinity of streets and crossroads (Figure 2, Figure 3 and Figure 4). The hedgehogs in our study seemed to avoid these spots of high light intensity when moving in their HRs, as all 22 hedgehogs exhibited lower animal_max values for their GPS locations compared to the random_max values of the simulated points. Additionally, the variance of the experienced light intensities (animal_SD) is, with the exception of three animals (TE_03_2019, TE_05_2019 and TE_07_2019), lower than the variance of the light intensities established through simulation of random locations within their HR (random_SD) (Table 2). These individuals were observed to have nests or movement paths situated along covering vegetation alongside illuminated areas, which might have enabled them to move relatively protected from ALAN sheltered by low bushes despite the presence of artificial light sources (corresponding to high ALAN values in the map).

Overall, a comparison of the intensity of ALAN experienced by the animals with the simulated random values indicated a preference for movement in areas with lower values of ALAN (Table 2). This behavioural pattern was observed regardless of the animal’s sex. This finding indicates a certain behavioural stability of ALAN avoidance in hedgehogs, which is interesting given that significant interindividual differences in movement behaviour of hedgehogs were observed in response to a music festival in a park in Berlin [48]. Despite the overall preference for movement in less strongly illuminated areas, comparison of movement behaviour in relation to ALAN between the three study locations indicated that hedgehogs living in TE differed in their behaviour compared to the hedgehogs living in TP and TR. Hedgehogs in TE showed a negative mean difference between simulated and experienced light intensities, meaning that they appear to prefer more strongly illuminated areas compared to the values obtained from a random distribution of locations in their home range. In TE, three hedgehogs exhibited a preference for more strongly illuminated areas compared to the simulated locations, which can be explained by the aforementioned locations of their nests and movement paths alongside roads and track beds, which display high ALAN intensities due to the streetlamps associated with these areas. However, since the light map used for the analysis maps light that was measured via overfly, these three hedgehogs might have actually experienced lower light intensities than those estimated in our analysis when moving through cover such as shrubs and bushes. This might explain the high light intensities estimated for those animals, which appear contradictory to the results of the remaining hedgehogs. However, it is also possible that these hedgehogs prefer more strongly illuminated areas due to other reasons, such as the avoidance of intraspecific competition for food resources or mating opportunities as well as other ecological factors correlating with ALAN intensity.

The preference for movement in areas with lower intensities of ALAN, which has been observed in wild hedgehogs in this study for the first time, is an important finding, since the response of nocturnal insectivores such as the hedgehog to ALAN might vary. First, they might intentionally seek out artificially illuminated areas to increase their nutritional intake. ALAN has the potential to attract invertebrates or affect their community composition [35,49], potentially leading to an altered or increased presence of predators feeding on invertebrates at artificially illuminated sites. This increased availability of prey has been suggested as a potential factor drawing hedgehogs into artificially illuminated areas such as roads [50]. Second, hedgehogs might avoid artificially lit areas to reduce the risk of encountering humans or predators, which can be detrimental to survival [51]. Concordant with this hypothesis, the nocturnal beach mouse (*Peromyscus polionotus*) prefers food patches situated further away from artificial light sources [52]. Similarly, brown rats (*Rattus norvegicus*) avoid artificially lit areas. In contrast, predatory species such as the fox (*Vulpes vulpes*), the stoat (*Mustela erminea*), the polecat (*Mustela putorius*), and the weasel (*Mustela nivalis*) prefer illuminated areas [53]. Interestingly, the badger (*Meles meles*), intraguild predator to the hedgehog [54,55], avoids urban and recreational areas as well as roads [1]. Roads as well as urban and recreational areas are closely linked to human presence and are thus expected to feature high ALAN intensities and these areas have been shown to have a positive effect on hedgehog presence in a previous study [1]. Together with the observed negative effect of badger presence on hedgehog population density, it has been suggested that hedgehogs might thus prefer illuminated areas due to a decreased risk of predation by badgers [1]. However, our findings did not support attraction to illuminated areas. Nonetheless, since we neither measured prey abundance nor badger distribution in our study locations, we can only hypothesise what caused the light-averse movement behaviour of the hedgehogs in our study. Based on local citizen science projects conducted in Berlin using camera trap data, we are aware that during the time of the study there had been no eagle owls and only few badgers detected in our study areas [56]. Furthermore, we encountered foxes, martens and raccoons during our nightly fieldwork but have not encountered badgers in our study locations. Hence it is likely, that badger densities in Great Britain are higher compared to the ones in Germany and especially compared to our study locations in Berlin. The lower intraguild predator density might hence affect the behaviour of the hedgehogs observed in our study, leading to the absence of a preference for intensely illuminated areas [12]. Taking the missing data on badger distribution into account, we conclude that our results can be best explained by a risk-avoidance strategy which causes hedgehogs to prefer less intensely lit areas.

Previous studies on hedgehog spatial behaviour in relation to ALAN neither found evidence for a preference nor an avoidance of illuminated areas [29,53]. First, such an indiscriminate response can be caused by habituation to ALAN. With ALAN becoming more and more abundant [57], animals thriving in artificially illuminated areas might even be selected for decreasing light sensitivity [19]. Molenaar et al. [53] used experimentally installed streetlights at drainage ditches connecting upland habitat and incorporated a habituation phase to ensure animals got accustomed to the changes. However, it is unknown which additional ecological factors correlate with ALAN intensity and might even have a stronger effect on the movement behaviour of hedgehogs. In concordance with this, Molenaar et al. [53] stated that, in contrast to ALAN, vegetation height indeed had a significant impact on the movement of hedgehogs. However, other ecological factors, such as vegetation height, prey abundance, impervious surface, or traffic, that might correlate with ALAN and impact hedgehog movement were not measured in our study. Thus, we can only hypothesise that a preference for less strongly illuminated areas might have been observed in our study due to other factors correlating with ALAN, such as vegetation height or cover. This is supported by data collected for the two hedgehogs in TE that exhibited exceptionally high light intensity values for their locations but seemed to nest and move in vegetation covering them from ALAN. The potential correlation with and impact of other ecological factors apart from ALAN might also explain why we do not see habituation to ALAN in our study, even though the majority of the light sources in our study are expected to be permanently installed. Without knowing the distribution of these additional ecological factors and their link to ALAN, it is difficult to disentangle the interconnectedness of these factors and compare the results of our study with those of Molenaar et al. [53].

Another study [29] investigated the behaviour of hedgehogs in relation to ALAN at supplementary feeding stations. However, because of the supplementary feeding, the study’s experimental set-up does not reflect a natural distribution of food sources and might mask the natural movement behaviour of wild hedgehogs. As decisions related to foraging are based on a trade-off between the benefits of energy intake and the risk of decreasing fitness by jeopardising survival, the risk posed by ALAN might be outweighed by the high amount of energetically valuable nutrition provided at the feeding station. In contrast, our study investigated the movement behaviour of hedgehogs in a natural unaltered setting without the confounding factor of supplementary feeding.

Although our results provide evidence for avoidance of artificially illuminated areas in hedgehogs, there are limitations to their interpretation. As we investigated the movement behaviour of wild hedgehogs in an unaltered urban setting, there are several factors that could not be controlled or are unknown. First, the correlation between the intensity of ALAN and other environmental factors affecting hedgehog behaviour are unknown but likely to exist. Since artificial light sources serve the purpose of illuminating areas for human use and are thus frequented by humans more intensely than dark areas, factors such as human disturbance, traffic, amount of impervious surface and vegetation structure might be linked to ALAN intensity. Furthermore, the distribution of food sources, both natural and anthropogenic, might be linked to ALAN intensity and levels of human presence. Hence, these factors might even have a stronger impact on movement behaviour than ALAN itself. Assessing or controlling these environmental parameters in upcoming studies will help unravel the contribution of these factors to shaping hedgehog behaviour and will aid in evaluating the influence of ALAN on hedgehog movement. Another limitation of this study is the estimation of ALAN intensities. The light map used was obtained via flyover and thus maps the emitted light detectable from above. This means that the light intensities correspond to the amount of light emitted upwards, including direct upward-facing light beams and scattered light. As the beams of streetlights serve to illuminate pathways, they often face downwards. Hedgehogs might thus be exposed to light intensities differing from those indicated in this light map. Furthermore, the light intensities mapped are displayed on a relative scale, which impedes comparisons with light intensities emitted by streetlights as well as other studies.

The measurement acuity of the GPS loggers undoubtedly impacts the acuity of the results as well. The loggers were programmed to record the GPS position every 5 or 10 min. Hedgehogs, however, have been reported to move with speeds of 1–2 m/s during brief sprints [58]. Together with the measurement error of the logger itself, which is approximately 10 m, and the resolution of 1 × 1 m of the light map, this might cause the measured GPS locations and thus the light intensity mapped to not reflect the exact actual position and ALAN intensity for an animal. However, due to the maximum weight of the loggers, which is limited by the hedgehog’s body weight, we were not able to use larger batteries allowing for locational fixes obtained at higher frequencies in order to account for the measurement error with repeated measurements. Bootstrap-analysis showed that for most animals, the number of GPS locations measured clearly exceeded the numbers necessary to calculate a reliable HR. Thus, HR estimation was accurate, but the high number of GPS positions also caused the corresponding number of simulated random points to fill up most of the HR. Thus, the simulated values reflected a mean value of the HR’s ALAN intensity rather than a simulated random path. Repetitions of these experiments should thus aim at achieving more accurate GPS positions with smaller inter-measurement intervals and a shorter overall measurement period to limit the total number of GPS positions.

Our study is the first to provide evidence that the movement of hedgehogs is related to ALAN intensity. As the results of this study should reflect natural behaviour, the obtained knowledge could be used to cushion population declines by increasing survival rates through conservation. Hedgehogs could therefore be led around dangerous areas such as roads by building dark corridors using vegetation and reduced illumination through streetlights. Apart from the threat of being killed in traffic, hedgehog populations in urban areas might face genetic isolation [16,17,18]. Bridging parks with dark corridors could help to safely connect isolated populations from different parks and thus increase genetic diversity. In this regard, empirical studies examining the sensory capabilities of hedgehogs using streetlights of different wavelengths and intensities would be helpful. The results of these studies can help in establishing guidelines for intensity thresholds and properties of streetlights by policymakers. Nonetheless, further studies are needed to confirm the influence of ALAN on the movement of wild European hedgehogs. Additionally, the influence of other environmental factors on the movement behaviour needs to be disentangled from the effect of ALAN. Shedding a light on the cryptic lifestyle of the European hedgehog, an urban adaptor species in serious decline, can help establish effective conservation measures for the protection of this nocturnal insectivore.

## 5. Conclusions

The results of our study provide unique evidence for an influence of ALAN on the movement behaviour of hedgehogs inhabiting urban spaces. European hedgehogs preferred less intensely lit areas compared to the ALAN intensity obtained via simulation. Moreover, this behaviour was observed regardless of sex and for 17 out of 22 individuals. Further studies are needed to confirm the role of ALAN and to disentangle it from the potential effects of other environmental factors.

## Figures and Tables

**Figure 1 animals-10-01306-f001:**
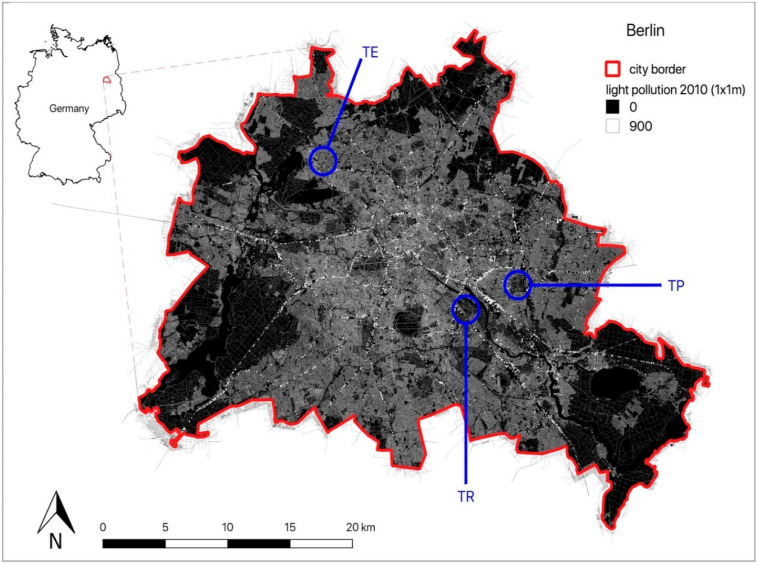
Map of Berlin showing the occurrence of artificial light at night. The three study areas, Tegel (TE), Treptower Park (TR) and Tierpark Berlin–Friedrichsfelde (TP), are marked. Light intensities are indicated by grey shades with increasing brightness in the map corresponding to increasing light intensities. Light intensity is indicated on a relative scale without a measurement unit.

**Figure 2 animals-10-01306-f002:**
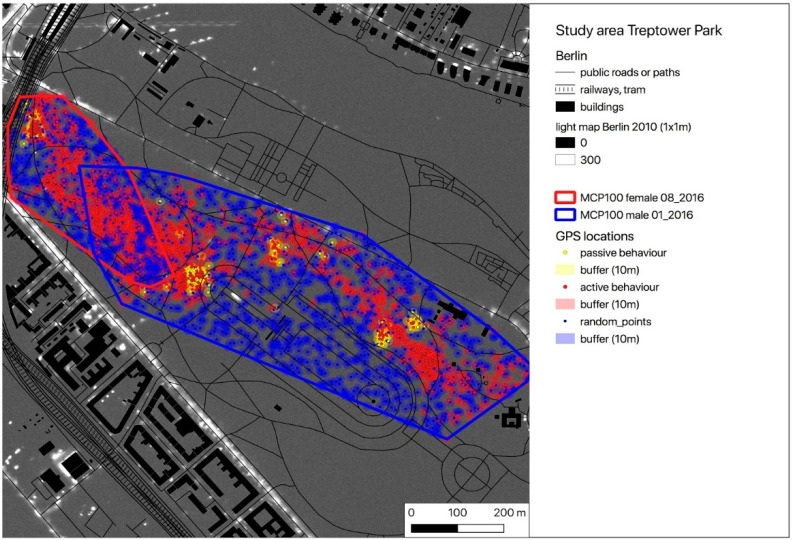
Map of the GPS locations of a female (TR_08_2016) and a male (TR_01_2016) hedgehog in the study area Treptower Park (yellow dots—locations with inactive behaviour = GPS_pas, red dots—locations with active behaviour = GPS_act). Blue dots represent randomly distributed locations within the individual Minimum Convex Polygon (MCP100) (GPS_random) (the number of GPS_act equals the number of GPS_random). All locations are buffered (radius = GPS error = 10 m). The overlayed light map of Berlin displays the occurrence and intensity of artificial light at night (ALAN).

**Figure 3 animals-10-01306-f003:**
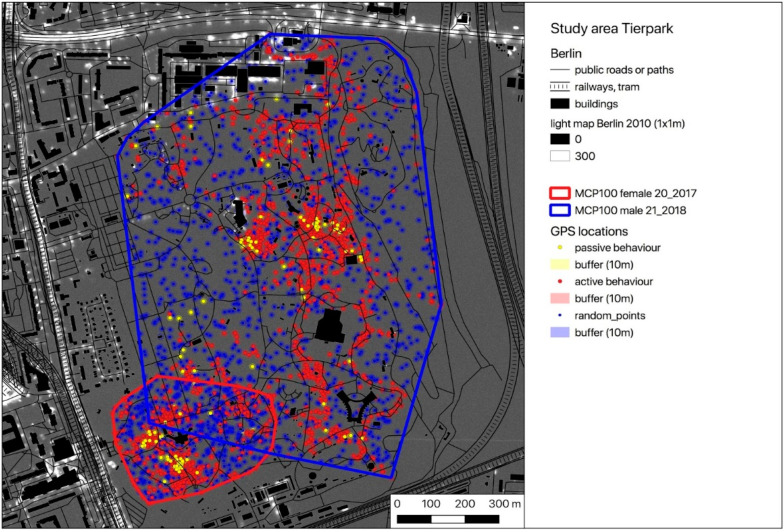
Map of the GPS locations of a female (TP_20_2017) and a male (TP_21_2018) hedgehog in the study area Tierpark (yellow dots—locations with inactive behaviour = GPS_pas, red dots— locations with active behaviour = GPS_act). Blue dots represent randomly distributed locations within the individual MCP100 (GPS_random) (the number of GPS_act equals the number of GPS_random). All locations are buffered (radius = GPS error = 10 m). The overlayed light map of Berlin shows the occurrence and intensity of ALAN.

**Figure 4 animals-10-01306-f004:**
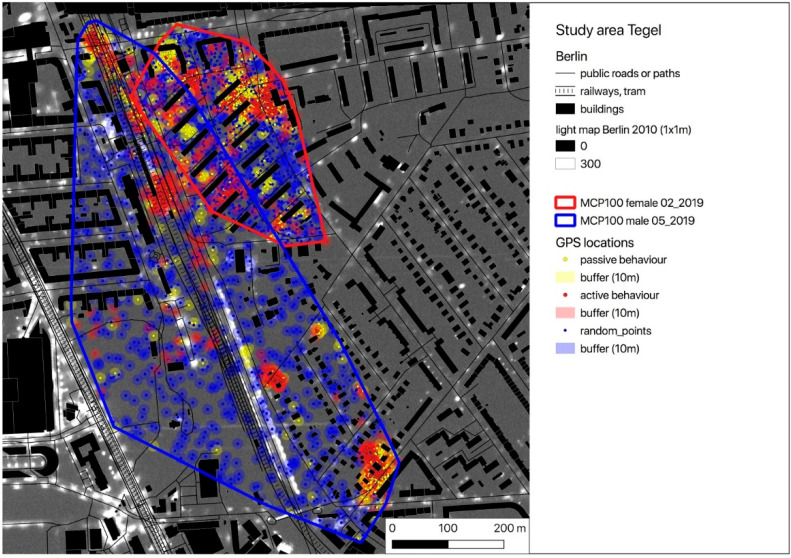
Map of the GPS locations of a female (TE_02_2019) and a male (TE_05_2019) hedgehog in the study area Tegel (yellow dots—locations with inactive behaviour = GPS_pas, red dots—locations with active behaviour = GPS_act). Blue dots represent randomly distributed locations within the individual MCP100 (GPS_random) (the number of GPS_act equals the number of GPS_random). All locations are buffered (radius = GPS error = 10 m). The overlayed light map of Berlin shows the occurrence and intensity of ALAN.

**Table 1 animals-10-01306-t001:** Overview of the study animals and data collected. Animal_ID: identification number including the study area (Tegel (TE), Treptower Park (TR) and Tierpark Berlin–Friedrichsfelde (TP)) and study year; sex: m = male, f = female; start date = tagging; end date = detagging; number of Global Positioning System (GPS)_total (total number of locations after cleaning), GPS_act (locations recorded during active behaviour), GPS_pas (locations recorded during passive behaviour); ACC_thres: activity threshold between active/passive behaviour; GPS interval: GPS measurement interval in minutes.

Animal_ID	Sex	Start Date [YYYY-MM-DD]	End Date [YYYY-MM-DD]	Number GPS_Total	Number GPS_Act	Number GPS_Pas	ACC_Thres	GPS Interval [Min]
TR_01_2016	m	2016-08-10	2016-09-19	1313	1140	173	1.87	10
TR_02_2016	f	2016-08-10	2016-09-19	1141	936	205	2.01	10
TR_08_2016	f	2016-08-10	2016-08-27	576	514	62	0.97	10
TR_09_2016	m	2016-08-10	2016-08-24	528	459	69	0.60	10
TR_13_2016	f	2016-08-10	2016-09-19	1357	1237	120	0.82	10
TR_17_2016	f	2016-08-10	2016-09-18	1227	1098	129	0.74	10
TR_19_2016	m	2016-08-10	2016-09-19	1275	1090	185	0.89	10
TR_21_2016	m	2016-08-10	2016-09-19	2566	2450	116	0.84	10
TP_13_2018	m	2018-05-22	2018-06-30	1127	1071	56	0.86	5
TP_18_2018	m	2018-05-22	2018-06-30	1110	994	116	0.99	5
TP_20_2017	f	2017-08-14	2017-09-04	489	449	40	0.84	10
TP_21_2018	m	2018-05-22	2018-06-30	1226	1151	75	0.84	5
TP_26_2017	f	2017-08-16	2017-09-12	487	400	87	0.88	10
TP_28_2017	f	2017-08-14	2017-09-04	684	606	78	0.92	10
TP_29_2018	m	2018-05-28	2018-06-23	941	828	113	0.94	5
TP_30_2018	m	2018-05-23	2018-06-09	991	838	153	0.94	5
TE_02_2019	f	2019-07-20	2019-09-30	973	446	527	1.15	5
TE_03_2019	m	2019-07-10	2019-08-01	870	633	237	1.00	5
TE_04_2019	m	2019-08-30	2019-10-08	1769	1100	669	0.86	5
TE_05_2019	m	2019-07-24	2019-09-16	1318	745	573	1.03	5
TE_07_2019	m	2019-07-24	2019-09-27	1906	1320	586	1.06	5
TE_10_2019	f	2019-08-24	2019-10-08	2469	1595	874	0.69	5

**Table 2 animals-10-01306-t002:** Mean light intensity values. Presented are the results (mean, SD = standard deviation, min = minimum, max = maximum) of the light intensity estimation for the buffered hedgehog’s GPS locations (animal_) and for the corresponding random points (random_). from the results of the statistical test for differences between animal_mean and random_mean, *p*-values and the significance level (0.05 = *, 0.01 = **, 0.001 = ***) are listed. Statistical test used: randomisation method.

Animal_ID	Animal_Mean	Animal_SD	Animal_Min	Animal_Max	Random_Mean	Random_SD	Random_Min	Random_Max	*p*-Values	Significance Level
TR_01_2016	102.0	4.2	99.3	140.9	118.9	25.1	101.4	226.8	0.001	***
TR_02_2016	102.8	6.2	99.6	177.0	125.2	150.5	98.6	10,339.3	0.001	***
TR_08_2016	106.5	20.2	100.6	290.2	125.6	159.2	99.0	8220.8	0.03	*
TR_09_2016	102.3	5.4	100.2	205.6	126.4	165.2	99.0	9845.4	0.001	***
TR_13_2016	115.2	39.6	100.2	578.1	125.8	160.3	98.7	11,076.4	0.993	n.s.
TR_17_2016	103.4	5.1	100.1	179.0	125.8	155.6	98.7	10,103.6	0.001	***
TR_19_2016	102.0	2.4	100.0	123.6	125.9	170.3	98.8	11,093.1	0.001	***
TR_21_2016	102.9	7.4	99.8	212.8	125.2	165.3	98.6	11,295.8	0.001	***
mean_TR	104.6	11.3	100.0	238.4	124.8	143.9	99.1	9025.1	-	-
TP_13_2018	105.2	26.9	100.5	544.2	120.6	83.9	43.6	3386.4	0.74	n.s.
TP_18_2018	102.7	1.1	100.7	116.3	120.5	81.8	47.8	3404.3	0.001	***
TP_20_2017	102.7	0.8	101.1	106.2	119.8	70.8	50.0	3269.8	0.001	***
TP_21_2018	103.9	9.9	100.6	235.7	120.2	80.4	44.8	3371.5	0.004	**
TP_26_2017	103.3	6.1	100.7	166.2	118.3	57.0	49.2	2835.1	0.001	***
TP_28_2017	103.0	1.2	100.1	106.6	120.5	86.4	43.9	3360.7	0.001	***
TP_29_2018	116.8	25.0	102.1	366.0	120.7	84.4	47.0	3317.2	0.987	n.s.
TP_30_2018	105.0	7.6	101.3	197.5	120.3	78.7	45.9	3337.0	0.005	**
mean_TP	105.3	9.8	100.9	229.8	120.1	77.9	46.5	3285.2	-	-
TE_02_2019	118.9	25.1	101.4	226.8	122.3	83.7	10.4	3348.0	0.001	***
TE_03_2019	159.8	351.7	101.5	3274.0	122.3	81.1	7.3	3332.2	1.0	n.s.
TE_04_2019	120.8	47.1	100.8	445.0	122.2	84.8	6.0	3346.3	0.02	*
TE_05_2019	160.9	86.3	101.9	443.3	122.2	82.6	5.1	3294.6	1.0	n.s.
TE_07_2019	127.0	44.4	101.2	623.6	122.3	83.2	6.1	3402.3	0.02	*
TE_10_2019	112.7	25.3	101.4	399.7	122.1	81.5	5.4	3396.6	0.001	***
mean_TE	133.4	96.6	101.4	902.1	122.2	82.8	6.7	3353.3	-	-

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
