# Peer review of "Moving in the Dark—Evidence for an Influence of Artificial Light at Night on the Movement Behaviour of European Hedgehogs (Erinaceus europaeus)"

_animals, 2020, doi:10.3390/ani10081306_

Round 1

Reviewer 1 Report

Dear authors, 

This is a very interesting paper and I am pleased to see research has been undertaken to understand more the impact of urbanisation on hedgehog behaviour. It is very interesting there isn't a difference between study sites, you might imagine that hedgehogs become adapted to their environments and more 'city' hedgehogs just adapt to increased light intensity, rather than continuing largely to avoid it. 

I enjoyed reading your manuscript, felt it was well written and clear. I have provided some very minor feedback below. I hope this is useful. 

Line 21: add in illuminated

Line 46: not sure on ‘apart of this’ – however?

Line 47: remove ‘even’

Line 83/84: move this into the methods

What area did you do your search over in TE?

Line 155/156: what do you mean data points recorded at times during which the data loggers were not attached to the animals? I would remove this section of the sentence as surely these aren’t data points?

Line 173/174: I am not clear by what you mean when you said you sorted all GPS locations from 7pm to 1hr after sunset and 1hr before sunrise until 7am into GPS_pas – I think this sentence might need reformatting to clarify

Is there maybe a link between light pollution and cover (so there is less cover in highly illuminated areas and it is the cover hedgehogs seek), and that’s why you are seeing less movement in well lit areas (e.g. line 332), rather than light per se?

Line 358: did you look at badger abundance? If not then you can’t say whether your hedgehogs were or were not avoiding badgers

Line 361: missing ‘in’ before Germany

Reviewer 2 Report

GENERAL COMMENTS

This is an interesting paper and brings useful and novel information for applied research on hedgehogs’ conservation and management.

However, a few methodological aspects should be further clarified and some parts of the text need to be re-structured (see specific comments below).

The abstract could be more informative. A sentence like “We were able to show that hedgehogs prefer to move in darker areas, regardless of sex, study area and season.” is ok for a Simple summary, but in the abstract it would be interesting to find more specific information, e.g. about significant differences that confirm the preference of hedgehogs for darker areas.

Figures and tables showing the research results should not be placed in the discussion, but in the results section, and should possibly be placed in the text close to where they are mentioned.

The most important general remark is that authors should keep in mind that in the discussion they should only discuss the results presented in the results section, without adding novel information, and without referring to data that have not been presented in the results (see specific comments below).

Unfortunately, the authors did not use line numbers, as required by the journal guidelines, and therefore it is more difficult to indicate the precise location of the changes required. I kindly ask the authors to provide line numbers in the new revised version.

SPECIFIC COMMENTS

Please be consistent in the way you write time (e.g. 07pm or 7pm?) or other items (e.g. MCP100 or MCP100%?). Check throughout the text.

page 4 (bottom): the sentence “As hedgehogs are strongly nocturnal, we sorted all GPS locations from 7pm to 1 hour after sunset and 1 hour before sunrise until 7am into GPS_pas.” is not clear. How did you know that they were passive 1 hour before sunset and 1 hour after sunrise? A few lines below, authors also say that “Locations during the active phase of the hedgehogs (1 hour after sunset to 1 hour before sunrise),…”. This seems in contrast with the previous sentence. Also, wasn’t activity level defined on the basis of accelerometers data? Or did you define it “a priori”, based on the time of the day (page 6: “According to the individual activity thresholds (ACC_thres), the GPS locations were divided into locations at times of inactive behaviour (GPS_pas) and times of active behaviour (GPS_act)”)? This is quite confusing…

page 5: “In the light map, light intensities are mapped using a relative scale from 0 (absolute darkness) to 48,429 (highest light intensity mapped).” What is the measurement unit? Authors mention that there is no measurement unit. But then what is 48,429? How was this value obtained? What and how did you measure to get it?

page 5: “period of 14-72 days”: please clarify to which period and to which days you refer to. This is not clear from the description of methodology nor from the table. If these days refer to the same days mentioned a few lines below (“all others we examined over a period of 21 days or longer”), then please connect these two sentences and avoid interrupting with other information. Use a clearer structure of the text.

page 6 and 10: please check table captions (they end below or at the side of the table). In Table 1 also specify date format (YYY-MM-DD). In Table 2 please also add “n.s.” in the column “significance level” when differences are not significant.

page 7: the whole initial paragraph (from “Human” to “urban setting”) is just a sort of abstract or repetition of concepts already presented in the text. It does not really present any discussion related to the data presented above, therefore it should be removed.

page 10: “The GPS-locations obtained from 9 female and 13 male European hedgehogs all displayed the animals' site fidelities (Figure S5-S8), allowing the estimation of HR calculation [43, 44]. Bootstrap analysis demonstrated that the amount of measured GPS-locations is sufficient for HR estimation (Figure S1-S4).” This seems to be more appropriate for the Methods section rather than for the discussion.

page 10: “For the animals that did not show a significant difference between the measured and random light intensities, we observed that their nest or their movement path was situated along small bushes directly bordering the roadside or the track bed.” This should have been presented in the results.

page 10: In the discussion the authors say that “This behavioural pattern was observed in all three parks regardless of the animal's sex, season, length of measurement interval, and year of measurement”. However, in the results section I could not find any analysis to support this statement. In the results please mention all the analysis carried out to check for differences depending on sex, season (which were the seasons tested? How were they divided?), etc. The lack of significance of these factors is also mentioned in the abstract and in the conclusions, but again you cannot mention it if you did not show any evidence of it.

page 11: “These areas, which are closely linked to human presence and are thus expected to feature high ALAN intensities…”. When you say “these areas”, to which areas are you exactly referring to? In the previous stamen you talk about “urban and recreational areas as well as roads”. So do you refer to all these areas? Or only some of these areas? Please specify. You also say that these areas “had a positive effect on hedgehog presence”: in which sense? Does this derive from results from other studies?

page 11: How can you say that “our findings did neither support this hypothesis of badger avoidance”? To state this, you should have data about badger distribution in your study areas. Do you have them? I could not see results about hedgehog presence in relation with badger presence in the manuscript…

page 11: “Previous studies on hedgehog spatial behaviour in relation to ALAN neither found evidence for a preference nor an avoidance of illuminated areas in hedgehogs [64, 29]. First, such an indiscriminate response can be caused by habituation to ALAN. With ALAN becoming more and more abundant [67], animals thriving in artificially illuminated areas might even be selected for decreasing light sensitivity [19]. However, as Molenaar et al., [64] used experimentally installed streetlights at drainage ditches connecting upland habitat, we assume that it is unlikely that the animals were habituated to ALAN.” This seems contradictory: no habituation can be supposed in the study by Molenaar, but they did not find avoidance reactions to ALAN, whereas hedgehogs in your study should be habituated to ALAN, yet they show ALAN avoidance responses… How do you explain this?

Round 2

Reviewer 2 Report

I appreciate the effort made by the authors to provide detailed answers to each of my comments.

I think that this new improved version is almnost ready to be accepted for publication.

As a minor comment, please note that Table 1 and Table 2 are repeated twice, with different formats (the two Table 2 are almost overlapping). Please check.

Author Response

Dear Ms. Wang, dear reviewer, 

We are very grateful to the reviewer for the effort put in reviewing our manuscript and for her/his thoughtful comments on it.

We have checked the latest version of our manuscript and assume that the issue with the duplicated tables is most likely caused by the tracked-changes function of Windows‘ Word. When we hid our changes made to the manuscript by clicking on the red ribbon on the left side of the text, the old versions of the tables were not visible any more. When we let Word display the changes by clicking again, then both the old and the new version of the table were shown because we had replaced the old version with the new one at this point in the manuscript and Word tracks these changes. The duplicated tables were also not visible in the PDF file that we submitted last time, in which we accepted all changes prior to establishing the PDF file.

We now submitted a Word file of our manuscript in which we accepted the changes to show that there is only the latest version of the tables visible. We have indicated the lines where the tables appear. We hope that our paper is now acceptable for publication and look forward to hearing from you in due course.

Best wishes,

Nadine Schubert (on behalf of all co-authors)

This manuscript is a resubmission of an earlier submission. The following is a list of the peer review reports and author responses from that submission.